# Local Causal Discovery of Direct Causes and Effects

**Tian Gao**      **Qiang Ji**
Department of ECSE
Rensselaer Polytechnic Institute, Troy, NY 12180
{gaot, jiq}@rpi.edu

## Abstract

We focus on the discovery and identification of direct causes and effects of a target variable in a causal network. State-of-the-art causal learning algorithms generally need to find the global causal structures in the form of complete partial directed acyclic graphs (CPDAG) in order to identify direct causes and effects of a target variable. While these algorithms are effective, it is often unnecessary and wasteful to find the global structures when we are only interested in the local structure of one target variable (such as class labels). We propose a new local causal discovery algorithm, called Causal Markov Blanket (CMB), to identify the direct causes and effects of a target variable based on Markov Blanket Discovery. CMB is designed to conduct causal discovery among multiple variables, but focuses only on finding causal relationships between a specific target variable and other variables. Under standard assumptions, we show both theoretically and experimentally that the proposed local causal discovery algorithm can obtain the comparable identification accuracy as global methods but significantly improve their efficiency, often by more than one order of magnitude.

## 1 Introduction

Causal discovery is the process to identify the causal relationships among a set of random variables. It not only can aid predictions and classifications like feature selection [4], but can also help predict consequences of some given actions, facilitate counter-factual inference, and help explain the underlying mechanisms of the data [13]. A lot of research efforts have been focused on predicting causality from observational data [13, 18]. They can be roughly divided into two sub-areas: causal discovery between a pair of variables and among multiple variables. We focus on multivariate causal discovery, which searches for correlations and dependencies among variables in causal networks [13]. Causal networks can be used for local or global causal prediction, and thus they can be learned locally and globally. Many causal discovery algorithms for causal networks have been proposed, and the majority of them belong to global learning algorithms as they seek to learn global causal structures. The Spirtes-Glymour-Scheines (SGS) [18] and Peter-Clark (P-C) algorithm [19] test for the existence of edges between every pair of nodes in order to first find the skeleton, or undirected edges, of causal networks and then discover all the V-structures, resulting in a partially directed acyclic graph (PDAG). The last step of these algorithms is then to orient the rest of edges as much as possible using Meek rules [10] while maintaining consistency with the existing edges. Given a causal network, causal relationships among variables can be directly read off the structure.

Due to the complexity of the P-C algorithm and unreliable high order conditional independence tests [9], several works [23, 15] have incorporated the Markov Blanket (MB) discovery into the causal discovery with a local-to-global approach. Growth and Shrink (GS) [9] algorithm uses the MBs of each node to build the skeleton of a causal network, discover all the V-structures, and then use the Meek rules to complete the global causal structure. The max-min hill climbing (MMHC) [23] algorithm also finds MBs of each variable first, but then uses the MBs as constraints to reduce the search space for the score-based standard hill climbing structure learning methods. In [15], authors

use Markov Blanket with Collider Sets (CS) to improve the efficiency of the GS algorithm by combining the spouse and V-structure discovery. All these local-to-global methods rely on the global structure to find the causal relationships and require finding the MBs for all nodes in a graph, even if the interest is the causal relationships between one target variable and other variables. Different MB discovery algorithms can be used and they can be divided into two different approaches: non-topology-based and topology-based. Non-topology-based methods [5, 9], used by CS and GS algorithms, greedily test the independence between each variable and the target by directly using the definition of Markov Blanket. In contrast, more recent topology-based methods [22, 1, 11] aim to improve the data efficiency while maintaining a reasonable time complexity by finding the parents and children (PC) set first and then the spouses to complete the MB.

Local learning of causal networks generally aims to identify a subset of causal edges in a causal network. Local Causal Discovery (LCD) algorithm and its variants [3, 17, 7] aim to find causal edges by testing the dependence/independence relationships among every four-variable set in a causal network. Bayesian Local Causal Discovery (BLCD) [8] explores the Y-structures among MB nodes to infer causal edges [6]. While LCD/BLCD algorithms aim to identify a subset of causal edges via special structures among all variables, we focus on finding all the causal edges adjacent to one target variable. In other words, we want to find the causal identities of each node, in terms of direct causes and effects, with respect to one target node. We first use Markov Blankets to find the direct causes and effects, and then propose a new Causal Markov Blanket (CMB) discovery algorithm, which determines the exact causal identities of MB nodes of a target node by tracking their conditional independence changes, without finding the global causal structure of a causal network. The proposed CMB algorithm is a complete local discovery algorithm and can identify the same direct causes and effects for a target variable as global methods under standard assumptions. CMB is more scalable than global methods, more efficient than local-to-global methods, and is complete in identifying direct causes and effects of one target while other local methods are not.

## 2  Backgrounds

We use $\mathcal{V}$ to represent the variable space, capital letters (such as $X, Y$) to represent variables, bold letters (such as $\mathbf{Z}, \mathbf{MB}$) to represent variable sets, and use $|\mathbf{Z}|$ to represent the size of set $\mathbf{Z}$. $X \perp\!\!\!\perp Y$ and $X \not\perp\!\!\!\perp Y$ represent independence and dependence between $X$ and $Y$, respectively. We assume readers are familar with related concepts in causal network learning, and only review a few major ones here. In a causal network or causal Bayesian Network [13], nodes correspond to the random variables in a variable set $\mathcal{V}$. Two nodes are adjacent if they are connected by an edge. A directed edge from node $X$ to node $Y$, $(X, Y) \in \mathcal{V}$, indicates $X$ is a *parent* or direct cause of $Y$ and $Y$ is a *child* or direct effect of $X$ [12]. Moreover, If there is a directed path from $X$ to $Y$, then $X$ is an *ancestor* of $Y$ and $Y$ is a *descendant* of $X$. If nonadjacent $X$ and $Y$ have a common child, $X$ and $Y$ are *spouses*. Three nodes $X, Y$, and $Z$ form a *V-structure* [12] if $Y$ has two incoming edges from $X$ and $Z$, forming $X \rightarrow Y \leftarrow Z$, and $X$ is not adjacent to $Z$. $Y$ is a *collider* in a path if $Y$ has two incoming edges in this path. Y with nonadjacent parents $X$ and $Z$ is an *unshielded* collider. A path $J$ from node $X$ and $Y$ is *blocked* [12] by a set of nodes $\mathbf{Z}$, if any of following holds true: 1) there is a non-collider node in $J$ belonging to $\mathbf{Z}$. 2) there is a collider node $C$ on $J$ such that neither $C$ nor any of its descendants belong to $\mathbf{Z}$. Otherwise, $J$ is unblocked or active.

A PDAG is a graph which may have both undirected and directed edges and has at most one edge between any pair of nodes [10]. CPDAGs [2] represent Markov equivalence classes of DAGs, capturing the same conditional independence relationships with the same skeleton but potentially different edge orientations. CPDAGs contain directed edges that has the same orientation for every DAG in the equivalent class and undirected edges that have reversible orientations in the equivalent class. Let $G$ be the causal DAG of a causal network with variable set $\mathcal{V}$ and $P$ be the joint probability distribution over variables in $\mathcal{V}$. $G$ and $P$ satisfy *Causal Markov condition* [13] if and only if, $\forall X \in \mathcal{V}$, $X$ is independent of non-effects of $X$ given its direct causes. The *causal faithfulness condition* [13] states that $G$ and $P$ are faithful to each other, if all and every independence and conditional independence entailed by $P$ is present in $G$. It enables the recovery of $G$ from sampled data of $P$. Another widely-used assumption by existing causal discovery algorithms is causal sufficiency [12]. A set of variables $\mathbf{X} \subseteq \mathcal{V}$ is *causally sufficient*, if no set of two or more variables in $\mathbf{X}$ shares a common cause variable outside $\mathcal{V}$. Without causal sufficiency assumption, latent confounders between adjacent nodes would be modeled by bi-directed edges [24]. We also assume *no selection bias* [20] and

we can capture the same independence relationships among variables from the sampled data as the ones from the entire population.

Many concepts and properties of a DAG hold in causal networks, such as d-separation and MB. A *Markov Blanket* [12] of a target variable $T$, $\mathbf{MB}_T$, in a causal network is the minimal set of nodes conditioned on which all other nodes are independent of $T$, denoted as $X \perp\!\!\!\perp T | \mathbf{MB}_T, \forall X \subseteq \{\mathbf{V} \setminus T\} \setminus \mathbf{MB}_T$. Given an unknown distribution $P$ that satisfied the Markov condition with respect to an unknown DAG $G^0$, Markov Blanket Discovery is the process used to estimate the MB of a target node in $G^0$, from independently and identically distributed (i.i.d) data $D$ of $P$. Under the causal faithfulness assumption between $G^0$ and $P$, the MB of a target node $T$ is unique and is the set of parents, children, and spouses of $T$ (i.e., other parents of children of $T$) [12]. In addition, the parents and children set of $T$, $\mathbf{PC}_T$, is also unique. Intuitively, the MB can directly facilitate causal discovery. If conditioning on the MB of a target variable $T$ renders a variable $X$ independent of $T$, then $X$ cannot be a direct cause or effect of $T$. From the local causal discovery point of view, although MB may contain nodes with different causal relationships with the target, it is reasonable to believe that we can identify their relationships exactly, up to the Markov equivalence, with further tests.

Lastly, exiting causal network learning algorithms all use three Meek rules [10], which we assume the readers are familiar with, to orient as many edges as possible given all V-structures in PDAGs to obtain CPDAG. The basic idea is to orient the edges so that 1) the edge directions do not introduce new V-structures, 2) preserve the no-cycle property of a DAG, and 3) enforce 3-fork V-structures.

## 3    Local Causal Discovery of Direct Causes and Effects

Existing MB discovery algorithms do not directly offer the exact causal identities of the learned MB nodes of a target. Although the topology-based methods can find the PC set of the target within the MB set, they can only provide the causal identities of some children and spouses that form v-structures. Nevertheless, following existing works [4, 15], under standard assumptions, every PC variable of a target can only be its direct cause or effect:

**Theorem 1.** *Causality within a MB. Under the causal faithfulness, sufficiency, correct independence tests, and no selection bias assumptions, the parent and child nodes within a target's MB set in a causal network contains all and only the direct causes and effects of the target variable.*

The proof can be directly derived from the PC set definition of a causal network. Therefore, using the topology-based MB discovery methods, if we can discover the exact causal identities of the PC nodes within the MB, causal discovery of direct causes and effects of the target can therefore be successfully accomplished.

Building on MB discovery, we propose a new local causal discovery algorithm, Causal Markov Blanket (CMB) discovery as shown in Algorithm 1. It identifies the direct causes and effects of a target variable without the need of finding the global structure or the MBs of all other variables in a causal network. CMB has three major steps: 1) to find the MB set of the target and to identify some direct causes and effects by tracking the independence relationship changes among a target's PC nodes before and after conditioning on the target node, 2) to repeat Step 1 but conditioned on one PC node's MB set, and 3) to repeat Step 1 and 2 with unidentified neighboring nodes as new targets to identify more direct causes and effects of the original target.

**Step 1: Initial identification**. CMB first finds the MB nodes of a target $T$, $\mathbf{MB}_T$, using a topology-based MB discovery algorithm that also finds $\mathbf{PC}_T$. CMB then uses the CausalSearch subroutine, shown in Algorithm 2, to get an initial causal identities of variables in $\mathbf{PC}_T$ by checking every variable pair in $\mathbf{PC}_T$ according to Lemma 1.

**Lemma 1.** *Let $(X, Y) \in \mathbf{PC}_T$, the PC set of the target $T \in \mathcal{V}$ in a causal DAG. The independence relationships between $X$ and $Y$ can be divided into the following four conditions:*

*C1 $X \perp\!\!\!\perp Y$ and $X \perp\!\!\!\perp Y | T$; this condition can not happen.*

*C2 $X \perp\!\!\!\perp Y$ and $X \not\perp\!\!\!\perp Y | T \Rightarrow X$ and $Y$ are both the parents of $T$.*

*C3 $X \not\perp\!\!\!\perp Y$ and $X \perp\!\!\!\perp Y | T \Rightarrow$ at least one of $X$ and $Y$ is a child of $T$.*

*C4 $X \not\perp\!\!\!\perp Y$ and $X \not\perp\!\!\!\perp Y | T \Rightarrow$ their identities are inconclusive and need further tests.*

---

**Algorithm 1** Causal Markov Blanket Discovery Algorithm

---

1: **Input:** $\mathcal{D}$: Data; $T$: target variable
2: **Output:** $ID_T$: the causal identities of all nodes with respect to $T$
   {Step 1: Establish initial ID }
3: $ID_T = zeros(|\mathcal{V}|, 1)$;
4: $(\mathbf{MB}_T, \mathbf{PC}_T) \leftarrow FindMB(T, \mathcal{D})$;
5: $\mathbf{Z} \leftarrow \emptyset$;
6: $ID_T \leftarrow CausalSearch(D, T, \mathbf{PC}_T, \mathbf{Z}, ID_T)$
   {Step 2: Further test variables with $id_T = 4$}
7: **for** one $X$ in each pair $(X, Y)$ with $id_T = 4$ **do**
8:     $\mathbf{MB}_X \leftarrow FindMB(X, \mathcal{D})$;
9:     $\mathbf{Z} \leftarrow \{\mathbf{MB}_X \setminus T\} \setminus Y$;
10:     $ID_T \leftarrow CausalSearch(D, T, \mathbf{PC}_T, \mathbf{Z}, ID_T)$;
11:     **if** no element of $ID_T$ is equal to 4, **break**;
12: **for** every pair of parents $(X, Y)$ of $T$ **do**
13:     **if** $\exists Z$ s.t. $(X, Z)$ and $(Y, Z)$ are $id_T = 4$ pairs **then**
14:         $ID_T(Z) = 1$;
15: $ID_T(X) \leftarrow 3, \forall X$ that $ID_T(X) = 4$;
   {Step 3: Resolve variable set with $id_T = 3$}
16: **for** each $X$ with $id_T = 3$ **do**
17:     Recursively find $ID_X$, without going back to the already queried variables;
18:     update $ID_T$ according to $ID_X$;
19:     **if** $ID_X(T) = 2$ **then**
20:         $ID_T(X) = 1$;
21:         **for** every $Y$ in $id_T = 3$ variable pairs $(X, Y)$ **do**
22:             $ID_T(Y) = 2$;
23:     **if** no element of $ID_T$ is equal to 3, **break**;
24: **Return:** $ID_T$

---

**Algorithm 2** CausalSearch Subroutine

---

1: **Input:** $\mathcal{D}$: Data; $T$: target variable; $\mathbf{PC}_T$: the PC set of $T$; $Z$: the conditioned variable set; $ID$: current ID
2: **Output:** $ID_T$: the new causal identities of all nodes with respect to $T$
   {Step 1: Single PC }
3: **if** $|\mathbf{PC}_T| = 1$ **then**
4:     $ID_T(\mathbf{PC}_T) \leftarrow 3$;
   {Step 2: Check C2 & C3}
5: **for** every $X, Y \in \mathbf{PC}_T$ **do**
6:     **if** $X \perp\!\!\!\perp Y | Z$ and $X \not\!\perp\!\!\!\perp Y | T \cup Z$ **then**
7:         $ID_T(X) \leftarrow 1; ID_T(Y) \leftarrow 1$;
8:     **else if** $X \not\!\perp\!\!\!\perp Y | Z$ and $X \perp\!\!\!\perp Y | T \cup Z$ **then**
9:         **if** $ID_T(X) = 1$ **then**
10:             $ID_T(Y) \leftarrow 2$
11:         **else if** $ID_T(Y) \neq 2$ **then**
12:             $ID_T(Y) \leftarrow 3$
13:     **if** $ID_T(Y) = 1$ **then**
14:         $ID_T(X) \leftarrow 2$
15:     **else if** $ID_T(X) \neq 2$ **then**
16:         $ID_T(X) \leftarrow 3$
17:     add $(X, Y)$ to pairs with $id_T = 3$
18:     **else**
19:         **if** $ID_T(X)$ & $ID_T(Y) = 0$ or 4 **then**
20:             $ID_T(X) \leftarrow 4; ID_T(Y) \leftarrow 4$
21:         add $(X, Y)$ to pairs with $id_T = 4$
   {Step 3: identify $id_T = 3$ pairs with known parents}
22: **for** every $X$ such that $ID_T(X) = 1$ **do**
23:     **for** every $Y$ in $id_T = 3$ variable pairs $(X, Y)$ **do**
24:         $ID_T(Y) \leftarrow 2$;
25: **Return:** $ID_T$

---

**C1** does not happen because the path $X - T - Y$ is unblocked either not given $T$ or given $T$, and the unblocked path makes $X$ and $Y$ dependent on each other. **C2** implies that $X$ and $Y$ form a V-structure with $T$ as the corresponding collider, such as node $C$ in Figure 1a which has two parents $A$ and $B$. **C3** indicates that the paths between $X$ and $Y$ are blocked conditioned on $T$, which means that either one of $(X, Y)$ is a child of $T$ and the other is a parent, or both of $(X, Y)$ are children of $T$. For example, node $D$ and $F$ in Figure 1a satisfy this condition with respect to $E$. **C4** shows that there may be another unblocked path from $X$ and $Y$ besides $X - T - Y$. For example, in Figure 1b, node $D$ and $C$ have multiple paths between them besides $D - T - C$. Further tests are needed to resolve this case.

Notation-wise, we use $ID_T$ to represent the causal identities for all the nodes with respect to $T$, $ID_T(X)$ as variable $X$'s causal identity to $T$, and the small case $id_T$ as the individual ID of a node to $T$. We also use $ID_X$ to represent the causal identities of nodes with respect to node $X$. To avoid changing the already identified PCs, CMB establishes a priority system[1]. We use the $id_T = 1$ to represent nodes as the parents of $T$, $id_T = 2$ children of $T$, $id_T = 3$ to represent a pair of nodes that cannot be both parents (and/or ambiguous pairs from Markov equivalent structures, to be discussed at Step 2), and $id_T = 4$ to represent the inconclusiveness. A lower number $id$ cannot be changed

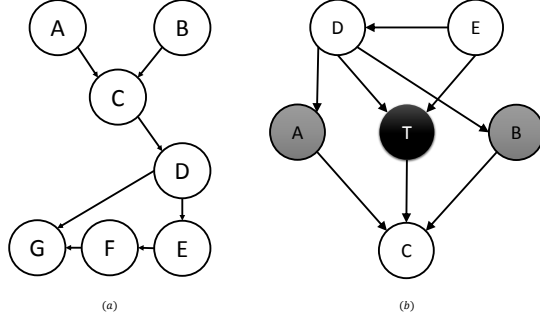

Figure 1: $a$) A Sample Causal Network. $b$) A Sample Network with C4 nodes. The only active path between $D$ and $C$ conditioned on $\mathbf{MB}_C \setminus \{T, D\}$ is $D - T - C$.

into a higher number (shown by Line 11~15 of Algorithm 2). If a variable pair satisfies **C2**, they will both be labeled as parents (Line 7 of Algorithm 2). If a variable pair satisfies **C3**, one of them is labeled as $id_T = 2$ only if the other variable within the pair is already identified as a parent; otherwise, they are both labeled as $id_T = 3$ (Line 9~12 and 15~17 of Algorithm 2). If a PC node remains inconclusive with $id_T = 0$, it is labeled as $id_T = 4$ in Line 20 of Algorithm 2. Note that if $T$ has only one PC node, it is labeled as $id_T = 3$ (Line 4 of Algorithm 2). Non-PC nodes always have $id_T = 0$.

**Step 2: Resolve $id_T = 4$.** Lemma 1 alone cannot identify the variable pairs in $\mathbf{PC}_T$ with $id_T = 4$ due to other possible unblocked paths, and we have to seek other information. Fortunately, by definition, the MB set of one of the target's PC node can block all paths to that PC node.

**Lemma 2.** *Let $(X, Y) \in \mathbf{PC}_T$, the PC set of the target $T \in \mathcal{V}$ in a causal DAG. The independence relationships between $X$ and $Y$, conditioned on the MB of $X$ minus $\{Y, T\}$, $\mathbf{MB}_X \setminus \{Y, T\}$, can be divided into the following four conditions:*

*C1 $X \perp\!\!\!\perp Y | \mathbf{MB}_X \setminus \{Y, T\}$ and $X \perp\!\!\!\perp Y | T \cup \mathbf{MB}_X \setminus Y$; this condition can not happen.*

*C2 $X \perp\!\!\!\perp Y | \mathbf{MB}_X \setminus \{Y, T\}$ and $X \not\!\perp\!\!\!\perp Y | T \cup \mathbf{MB}_X \setminus Y \Rightarrow X$ and $Y$ are both the parents of $T$.*

*C3 $X \not\!\perp\!\!\!\perp Y | \mathbf{MB}_X \setminus \{Y, T\}$ and $X \perp\!\!\!\perp Y | T \cup \mathbf{MB}_X \setminus Y \Rightarrow$ at least one of $X$ and $Y$ is a child of $T$.*

*C4 $X \not\!\perp\!\!\!\perp Y | \mathbf{MB}_X \setminus \{Y, T\}$ and $X \not\!\perp\!\!\!\perp Y | T \cup \mathbf{MB}_X \setminus Y \Rightarrow$ then $X$ and $Y$ is directly connected.*

**C1~3** are very similar to those in Lemma 1. **C4** is true because, conditioned on $T$ and the MB of $X$ minus $Y$, the only potentially unblocked paths between $X$ and $Y$ are $X - T - Y$ and/or $X - Y$. If **C4** happens, then the path $X - T - Y$ has no impact on the relationship between $X$ and $Y$, and hence $X - Y$ must be directly connected. If $X$ and $Y$ are not directly connected and the only potentially unblocked path between $X$ and $Y$ is $X - T - Y$, and $X$ and $Y$ will be identified by Line 10 of Algorithm 1 with $id_T \in \{1, 2, 3\}$. For example in Figure 1b, conditioned on $\mathbf{MB}_C \setminus \{T, D\}$, i.e., $\{A, B\}$, the only path between $C$ and $D$ is through T. However, if $X$ and $Y$ are directly connected, they will remain with $id_T = 4$ (such as node $D$ and $E$ from Figure 1b). In this case, $X$, $Y$, and $T$ form a fully connected clique, and edges among the variables that form a fully connected clique can have many different orientation combinations without affecting the conditional independence relationships. Therefore, this case needs further tests to ensure Meek rules are satisfied. The third Meek rule (enforcing 3-fork V-structures) is first enforced by Line 14 of Algorithm 1. Then the rest of $id_T = 4$ nodes are changed to have $id_T = 3$ by Line 15 of Algorithm 1 and to be further processed (even though they could be both parents at the same time) with neighbor nodes' causal identities. Therefore, Step 2 of Algorithm 1 makes all variable pairs with $id_T = 4$ to become identified either as parents, children, or with $id_T = 3$ after taking some neighbors' MBs into consideration. Note that Step 2 of CMB only needs to find the MB's for a small subset of the PC variables (in fact only one MB for each variable pair with $id_T = 4$).

**Step 3: Resolve $id_T = 3$.** After Step 2, some PC variables may still have $id_T = 3$. This could happen because of the existence of Markov equivalence structures. Below we show the condition under which the CMB can resolve the causal identities of all PC nodes.

**Lemma 3.** *The Identifiability Condition. For Algorithm 1 to fully identify all the causal relationships within the PC set of a target $T$, 1) $T$ must have at least two nonadjacent parents, 2) one of $T$'s single ancestors must contain at least two nonadjacent parents, or 3) $T$ has 3 parents that form a 3-fork pattern as defined in Meeks rules.*

We use single ancestors to represent ancestor nodes that do not have a spouse with a mutual child that is also an ancestor of $T$. If the target does not meet any of the conditions in Lemma 2, **C2** will never be satisfied and all PC variables within a MB will have $id_T = 3$. Without a single parent identified, it is impossible to infer the identities of children nodes using **C3**. Therefore, all the identities of the PC nodes are uncertain, even though the resulting structure could be a CPDAG.

Step 3 of CMB searches for a non-single ancestor of $T$ to infer the causal directions. For each node $X$ with $id_T = 3$, CMB tries to identify its local causal structure recursively. If $X$'s PC nodes are all identified, it would return to the target with the resolved identities; otherwise, it will continue to search for a non-single ancestor of $X$. Note that CMB will not go back to already-searched variables with unresolved PC nodes without providing new information. Step 3 of CMB checks the identifiability condition for all the ancestors of the target. If a graph structure does not meet the conditions of Lemma 3, the final $ID_T$ will contain some $id_T = 3$, which indicates reversible edges in CPDAGs. The found causal graph using CMB will be a PDAG after Step 2 of Algorithm 1, and it will be a CPDAG after Step 3 of Algorithm 1.

**Case Study**. The procedure using CMB to identify the direct causes and effects of $E$ in Figure 1a has the following 3 steps. **Step 1**: CMB finds the MB and PC set of $E$. The PC set contains node $D$ and $F$. Then, $ID_E(D) = 3$ and $ID_E(F) = 3$. **Step 2**: to resolve the variable pair $D$ and $F$ with $id_E = 3$, 1) CMB finds the PC set of $D$, containing $C$, $E$, and $G$. Their $id_D$ are all 3's, since $D$ contains only one parent. 2) To resolve $ID_D$, CMB checks causal identities of node $C$ and $G$ (without going back to $E$). The PC set of $C$ contains $A$, $B$, and $D$. CMB identifies $ID_C(A) = 1$, $ID_C(B) = 1$, and $ID_C(D) = 2$. Since $C$ resolves all its PC nodes, CMB returns to node $D$ with $ID_D(C) = 1$. 3) With the new parent $C$, $ID_D(G) = 2, ID_D(E) = 2$, and CMB returns to node $E$ with $ID_E(D) = 1$. **Step 3**: the $ID_E(D) = 1$, and after resolving the pair with $id_E = 3$, $ID_E(F) = 2$.

**Theorem 2.** *The Soundness and Completeness of CMB Algorithm. If the identifiability condition is satisfied, using a sound and complete MB discovery algorithm, CMB will identify the direct causes and effects of the target under the causal faithfulness, sufficiency, correct independence tests, and no selection bias assumptions.*

*Proof.* A sound and complete MB discovery algorithm find all and only the MB nodes of a target. Using it and under the causal sufficiency assumption, the learned PC set contains all and only the cause-effect variables by Theorem 1. When Lemma 3 is satisfied, all parent nodes are identifiable through V-structure independence changes, either by Lemma 1 or by Lemma 2. Also since children cannot be conditionally independent of another PC node given its MB minus the target node (**C2**), all parents identified by Lemma 1 and 2 will be the true positive direct causes. Therefore, all and only the true positive direct causes will be correctly identified by CMB. Since PC variables can only be direct causes or direct effects, all and only the direct effects are identified correctly by CMB. □

In the cases where CMB fails to identify all the PC nodes, global causal discovery methods cannot identify them either. Specifically, structures failing to satisfy Lemma 3 can have different orientations on some edges while preserving the skeleton and v-structures, hence leading to Markov equivalent structures. For the cases where $T$ has all single ancestors, the edge directions among all single ancestors can always be reversed without introducing new V-structures and DAG violations, in which cases the Meek rules cannot identify the causal directions either. For the cases with fully connected cliques, these fully connected cliques do not meet the nonadjacent-parents requirement for the first Meek rule (no new V-structures), and the second Meek rule (preserving DAGs) can always be satisfied within a clique by changing the direction of one edge. Since CMB orients the 3-fork V-structure in the third Meek rule correctly by Line 12~14 of Algorithm 1, CMB can identify the same structure as the global methods that use the Meek rules.

**Theorem 3.** *Consistency between CMB and Global Causal Discovery Methods. For the same DAG $G$, Algorithm 1 will correctly identify all the direct causes and effects of a target variable $T$*

*as the global and local-to-global causal discovery methods[2] that use the Meek rules [10], up to G's CPDAG under the causal faithfulness, sufficiency, correct independence tests, and no selection bias assumptions.*

*Proof.* It has been shown that causal methods using Meek rules [10] can identify up to a graph's CPDAG. Since Meek rules cannot identify the structures that fail Lemma 3, the global and local-to-global methods can only identify the same structures as CMB. Since CMB is sound and complete in identifying these structures by Theorem 2, CMB will identify all direct causes and effects up to G's CPDAG. □

### 3.1 Complexity

The complexity of CMB algorithm is dominated by the step of finding the MB, which can have an exponential complexity [1, 16]. All other steps of CMB are trivial in comparison. If we assume a uniform distribution on the neighbor sizes in a network with $N$ nodes, then the expected time complexity of Step 1 of CMB is $O(\frac{1}{N}\sum_{i=1}^{N} 2^i) = O(\frac{2^N}{N})$, while local-to-global methods are $O(2^N)$. In later steps, CMB also needs to find MBs for a small subset of nodes that include 1) one node between every pair of nodes that meet **C4**, and 2) a subset of the target's neighboring nodes that provide additional clues for the target. Let $l$ be the total size of these nodes, then CMB reduces the cost by $\frac{N}{l}$ times asymptotically.

## 4 Experiments

We use benchmark causal learning datasets to evaluate the accuracy and efficiency of CMB with four other causal discovery algorithms discussed: P-C, GS, MMHC, CS, and the local causal discovery algorithm LCD2 [7]. Due to page limit, we show the results of the causal algorithms on four medium-to-large datasets: ALARM, ALARM3, CHILD3, and INSUR3. They contain 37 to 111 nodes. We use 1000 data samples for all datasets. For each global or local-to-global algorithm, we find the global structure of a dataset and then extract causal identities of all nodes to a target node. CMB finds causal identities of every variable with respect to the target directly. We repeat the discovery process for each node in the datasets, and compare the discovered causal identities of all the algorithms to all the Markov equivalent structures with the known ground truth structure. We use the edge scores [15] to measure the number of missing edges, extra edges, and reversed edges[3] in each node's local causal structure and report average values along with its standard deviation, for all the nodes in a dataset. We use the existing implementation [21] of HITON-MB discovery algorithm to find the MB of a target variable for all the algorithms. We also use the existing implementations [21] for P-C, MMHC, and LCD2 algorithms. We implement GS, CS, and the proposed CMB algorithms in MATLAB on a machine with 2.66GHz CPU and 24GB memory. Following the existing protocol [15], we use the number of conditional independence tests needed (or scores computed for the score-based search method MMHC) to find the causal structures given the MBs[4], and the number of times that MB discovery algorithms are invoked to measure the efficiency of various algorithms. We also use mutual-information-based conditional independence tests with a standard significance level of 0.02 for all the datasets without worrying about parameter tuning.

As shown in Table 1, CMB consistently outperforms the global discovery algorithms on benchmark causal networks, and has comparable edge accuracy with local-to-global algorithms. Although CMB makes slightly more total edge errors in ALARM and ALARM3 datasets than CS, CMB is the best method on CHILD3 and INSUR3. Since LCD2 is an incomplete algorithm, it never finds extra or reversed edges but misses the most amount of edges. Efficiency-wise, CMB can achieve more than one order of magnitude speedup, sometimes two orders of magnitude as shown in CHILD3 and INSUR3, than the global methods. Compared to local-to-global methods, CMB also can achieve

Table 1: Performance of Various Causal Discovery Algorithms on Benchmark Networks

| Dataset | Method | Errors: Extra | Edges Missing | Reversed | Total | Efficiency No. Tests | No. MB |
|---|---|---|---|---|---|---|---|
| Alarm | P-C | 1.59±0.19 | 2.19±0.14 | 0.32±0.10 | 4.10±0.19 | 4.0e3±4.0e2 | - |
| | MMHC | 1.29±0.18 | 1.94±0.09 | 0.24±0.06 | 3.46±0.23 | 1.8e3±1.7e3 | 37±0 |
| | GS | 0.39±0.44 | 0.87±0.48 | 1.13±0.23 | 2.39±0.44 | 586.5±72.2 | 37±0 |
| | CS | 0.42±0.10 | 0.64±0.10 | 0.38±0.08 | **1.43±0.10** | 331.4±61.9 | 37±0 |
| | LCD2 | 0.00±0.00 | 2.49±0.00 | 0.00±0.0 | 2.49±0.00 | 1.4e3±0 | - |
| | CMB | 0.69±0.13 | 0.61±0.11 | 0.51±0.10 | 1.81±0.11 | **53.7±4.5** | **2.61 ± 0.12** |
| Alarm3 | P-C | 3.71±0.57 | 2.21±0.25 | 1.37±0.04 | 7.30±0.68 | 1.6e4±4.0e2 | - |
| | MMHC | 2.36±0.11 | 2.45±0.08 | 0.72±0.08 | 5.53±0.27 | 3.7e3±6.1e2 | 111 ± 0 |
| | GS | 1.24±0.23 | 1.41±0.05 | 0.99±0.14 | 3.64±0.13 | 2.1e3±1.2e2 | 111 ± 0 |
| | CS | 1.26±0.16 | 1.47±0.08 | 0.63±0.14 | **3.38±0.13** | 699.1±60.4 | 111±0 |
| | LCD2 | 0.00±0.00 | 3.85±0.00 | 0.00±0.0 | 3.85±0.00 | 1.2e4±0 | - |
| | CMB | 1.41±0.13 | 1.55±0.27 | 0.78±0.25 | 3.73±0.11 | **50.3±6.2** | **2.58 ± 0.09** |
| Child3 | P-C | 4.32±0.68 | 2.69±0.08 | 0.84±0.10 | 7.76±0.98 | 8.3e4±2.9e3 | - |
| | MMHC | 1.98±0.10 | 1.57±0.04 | 0.43±0.04 | 4.00±0.93 | 6.6e3±8.2e2 | 60 ±0 |
| | GS | 0.88±0.04 | 0.75±0.08 | 1.03±0.08 | 2.66±0.33 | 2.1e3±2.5e2 | 60±0 |
| | CS | 0.94±0.20 | 0.91±0.14 | 0.53±0.08 | 2.37±0.33 | 1.0e3±4.8e2 | 60± 0 |
| | LCD2 | 0.00±0.00 | 2.63±0.00 | 0.00±0.0 | 2.63±0.00 | 3.6e3±0 | - |
| | CMB | 0.92±0.12 | 0.84±0.16 | 0.60±0.10 | **2.36±0.31** | **78.2±15.2** | **2.53 ± 0.15** |
| Insur3 | P-C | 4.76±1.33 | 2.50±0.11 | 1.29±0.11 | 8.55±0.81 | 2.5e5±1.2e4 | - |
| | MMHC | 2.39±0.18 | 2.53±0.06 | 0.76±0.07 | 5.68±0.43 | 3.1e4±5.2e2 | 81 ± 0 |
| | GS | 1.94±0.06 | 1.44±0.05 | 1.19±0.10 | 4.57±0.33 | 4.5e4±2.2e3 | 81±0 |
| | CS | 1.92±0.08 | 1.56±0.06 | 0.89±0.09 | 4.37±0.23 | 2.6e4±3.9e3 | 81±0 |
| | LCD2 | 0.00±0.00 | 5.03±0.00 | 0.00±0.0 | 5.03±0.00 | 6.6e3±0 | - |
| | CMB | 1.72±0.07 | 1.39±0.06 | 1.19±0.05 | **4.30±0.21** | **159.8±38.5** | **2.46 ± 0.11** |

more than one order of speedup on ALARM3, CHILD3, and INSUR3. In addition, on these datasets, CMB only invokes MB discovery algorithms between 2 to 3 times, drastically reducing the MB calls of local-to-global algorithms. Since independence test comparison is unfair to LCD2 who does not use MB discovery or find moral graphs, we also compared time efficiency between LCD2 and CMB. CMB is 5 times faster on ALARM, 4 times faster on ALARM3 and CHILD3, and 8 times faster on INSUR3 than LCD2.

In practice, the performance of CMB depends on two factors: the accuracy of independence tests and MB discovery algorithms. First, independence tests may not always be accurate and could introduce errors while checking the four conditions of Lemma 1 and 2, especially under insufficient data samples. Secondly, causal discovery performance heavily depends on the performance of the MB discovery step, as the error could propagate to later steps of CMB. Improvements on both areas could further improve CMB's accuracy. Efficiency-wise, CMB's complexity can still be exponential and is dominated by the MB discovery phrase, and thus its worst case complexity could be the same as local-to-global approaches for some special structures.

## 5   Conclusion

We propose a new local causal discovery algorithm CMB. We show that CMB can identify the same causal structure as the global and local-to-global causal discovery algorithms with the same identification condition, but uses a fraction of the cost of the global and local-to-global approaches. We further prove the soundness and completeness of CMB. Experiments on benchmark datasets show the comparable accuracy and greatly improved efficiency of CMB for local causal discovery. Possible future works could study assumption relaxations, especially without the causal sufficiency assumption, such as by using a similar procedure as FCI algorithm and the improved CS algorithm [14] to handle latent variables in CMB.

## Footnotes

[1]Note that the identification number is slightly different from the condition number in Lemma 1.

[2]We specify the global and local-to-global causal methods to be P-C [19], GS [9] and CS [15].

[3]If an edge is reversible in the equivalent class of the original graph but are not in the equivalent class of the learned graph, it is considered as reversed edges as well.

[4]For global methods, it is the number of tests needed or scores computed given the moral graph of the global structure. For LCD2, it would be the total number of tests since it does not use moral graph or MBs.

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
