[Reviews · NeurIPS 2015]

Submitted by Assigned_Reviewer_1

The goal of the paper is to identify direct causes and effects of a given variable (target variable). The main contribution stands in that the authors achieve their goal without the need of first discovering the whole causal graph before restricting attention to the neighborhood of the target variable. Instead, they directly focus on discovering the local causal structure.

Existing algorithms can discover the Markov blanket (MB) of a target variable and a subset of it that consists of the set of parents and children (PC) of the target variable. The goal of this paper is to identify which of the nodes in PC are parents and which children of the target.

Significance & Originality:

I consider the contribution of the paper significant. It improves on efficiency and scalability compared to previous methods which need to perform the intermediate step of global causal discovery to achieve the same goal.

Clarity:

I think that lack of clarity and flow in several parts is a drawback of the paper. I found some descriptions confusing. In general, I would suggest more explicit explanations, motivating examples and high level explanations before going into the technical parts.

Specifically:

- Last paragraph of Introduction: the authors refer to Local Causal Discovery. It took me some time to realize that they refer to existing work and not to theirs, since this term coincides with the title of the paper. I would suggest changing the title to something more specific like "local causal discovery of direct causes and effects". Moreover, I find the name of the proposed method "Causal Markov Blanket" also too general and maybe misleading. The authors could probably consider a common name for both title and method involving "direct causes and effects".

- Section 3: Some high level explanations are necessary before going into the technical details. The authors could provide a short high level summary of all steps of the proposed method in the beginning of section 3.2.

- I could be missing something in Algorithm 2: as far as I understand, ID_T should contain the causal identities of *just* the PC set of T. But in step 1 it is mentioned "..the causal identities of *all* nodes.." and in Algorithm 2 the size of ID_T is |V| where V is the entire variable space and no just the PC.

- The output of Algorithm 2 is the ID_T. How is this converted to a CPDAG? For id_T 1 and 2 it is clear. For id_T =3 do you draw an indirect edge? Providing some visual examples of the output CPDAG of the Algorithm would be helpful.

- Elaborating more on why failing to meet Lemma 3 leads to Markov equivalent structures, could be useful.

- The authors could consider improving the flow in page 3.

- Some parts are not very specific/explicit: e.g.: * Sec. 4, first par.: "causal identities for a target node": isn't it "causal identities of nodes in the target's PC set"? * I understand what is a CPDAG but it would be helpful to define what it is meant by the "CPDAG of a target variable". Is it a subset of the original CPDAG containing T and the parents and children of it? *In the proof of Theorem 3: "CMB will identify up to a graph's CPDAG": is this again the subset? * Footnote 3 is not clear * Sec. 3.1: what does it mean "some V-structure children and spouses" ? * Sec 3.1: "PC set of a MB" : isn't it "PC set of a target's MB"? * The terms "local causal discovery of the target" and "local causal nodes of the target" could be rephrased

Quality:

I have not checked all the technical details but the paper seems to have a well-supported theoretical analysis followed by experimental results. The results show significant improvements in efficiency compared to global methods.

-Typos: * There are several typos in the references section. * Introduction: in term -> in terms *Sec. 3.1: "... no selection bias, every.." -> "...no selection bias, that every..." * Experiments: "along its standard deviation" -> "along with its...", "As shown in in Table"-> remove one "in"

Summary: I consider the paper having a significant contribution in more efficiently finding direct causes and effects of a target variable. I found the presentation sometimes confusing, it could be improved by more visual examples, motivation and more clear explanations.

Submitted by Assigned_Reviewer_2

Weak accept

The description of the contributions should be clearer in the introduction. Your contributions relative to LCD and BLCD are not as clear as they should be.

The paper presents a new local search algorithm for graphical relationships. Under suitable assumptions these relationships are causal. The ideas in the local search algorithm are technically interesting.

I'm not convinced that comparing the efficiency to global search algorithms is fair or interesting.
Summary: light review

Submitted by Assigned_Reviewer_3

This paper considers the task of learning the direct causes and effects of a predefined target variable T, instead of the entire causal structure. The algorithm introduced in the paper departs from the Markov blanket, and builds on a few rules that apply over the neighborhood of T so as to discover the directionality of the arrows. It is a clean and interesting paper, and the algorithm is shown to be equivalent to previously established ones (i.e., doesn't miss anything). In fact, the algorithm does not discover any additional arrow than already known results, but it improves on the scalability since only a subset of the variables is the target of the analysis in many real-world scenarios, not the whole graph.
Summary: This paper considers the problem of learning the direct causes and effects of a predefined target variable T, instead of the entire causal structure.

Submitted by Assigned_Reviewer_4

This paper concerns learning direct causes and effects of a target variables using local learning. This paper extends existing local learning methods by taking account larger neighbourhoods and the presented method is therefore able to infer more causal relations than the previous methods. The problem is relevant and the contributions are worthwhile.

Overall, technical presentation is sloppy.

What is the definition of an "unshielded parent"? What is the definition of a "set of three-fork parents"? What is the definition of a "sound and complete MB discovery algorithm"?

Theorems 2 and 3 do not contain all necessary assumptions. In the proof of Theorem 2, causal sufficiency is used. However, it was not assumed in the theorem. One also need to

assume that conditional independence tests return always correct results (or infinite data or some other similar assumption). Also it is unclear to me why it isn't necessary that the distribution is faithful to any DAG?

099: Markov condition tells that a node is independent of its non-descendants given its parents, Causal Markov condition tells that a node is independent of its non-effects given its direct causes.

125, 227: R\(S\T) is usually different than (R\S)\T, so if you have two set differences in a row it is good to use parenthesis to avoid any misinterpretations.

133-135: Why is this a reasonable belief? We know that, if one uses only conditional independence tests, structures can be identified only up to a Markov equivalence class.

293: the PC set of a MB -> the PC set of a target

The experiments could be strengthened by adding a standard Markov blanket discovery algorithm (where you learn only the directions of arcs which are part of v-structures) as a baseline to see how much the new rules help.

The evaluation of the efficiency of different algorithms seems to give an unfair advantage to the CMB algorithm. The efficiency is measured in terms of the number of conditional tests and the number of MB discovery algorithms are invoked. The problem is that MB discovery algorithms typically use conditional independence tests to find MBs and not counting the conditional independence tests inside a call of an MB discovery algorithm favours algorithms that use such subroutines. One thing that could make evaluation more fair would be to add also the actual running times of different algorithms.
Summary: An interesting topic and a worthy contribution. However, technical representation should be improved.

Author Feedback
Author rebuttal: We appreciate reviewers' recognition of the contributions of this work. We will address each issue carefully and improve the paper's clarity and presentation. Below are our responses to reviewers' major concerns:

R1:
1. Title and algorithm name: We will change the paper title to "local causal discovery of direct causes and effects". "Causal Markov Blanket" is used to emphasize the discovery of direct causes and effects within the identified MB. A more specific algorithm name could be "Markov-Blanket-based direct causes and effects discovery"(MBDCED), but it seems long and redundant.
2. Clarity: we will include a high level summary of our algorithm in the beginning of Section 3 before discussing the details. We will also improve the flow of page 3.
3. Algorithm details: ID_T (or ID_X) is an indicator vector for all variables in V. For the non-PC variables, their id would just be zero. Moreover, the undetermined id_T = 3 represents undirected edges in a CPDAG. We will also provide an example with figures to better explain our algorithm.
4. Lemma 3: structures failing to meet Lemma 3 can have different orientations on some edges while preserving the skeleton and v-structures, hence leading to Markov equivalent structures. We will further clarify this statement.
5. Terminology: "CPDAG of a target variable" should be the subgraph of the global CPDAG containing the target and its PC set. "V-structure children and spouses" means the children and spouses that form v-structures. "PC set of a MB" means the PC set within the target's MB set. "Local causal nodes of the target" represents direct causes and effects of the target, and "local causal discovery of the target" is the discovery of these direct causes and effects. We will clarify above statements and fix other typos.

R2: thanks for the positive comments.

R3:
1. Presentation: we will significantly improve the clarity and presentation of this paper and define each term clearly before their first usage. "Unshielded parents" means the nonadjacent parents of a node, as per the definition of unshielded colliders. "Set of three-fork parents" means 3 parents of a node that form a 3-fork pattern as defined in Meek's rules. "Sound and complete MB discovery algorithms" find all and only the MB nodes of a target. We should use "non-effects" and "causes" in "Causal Markov condition" and will correct it.
2. Assumptions for theorems: Consistent with MB discovery algorithms and other causal discovery algorithms, our theorems require the standard faithfulness, correct independence tests, and causal sufficiency assumptions. We will explicitly state these assumptions in the theorems.
3. Line 125, 227: We will add the parenthesis to avoid misinterpretation.
4. Line 133-135: we wanted to introduce the main idea first without details. A more precise statement should be "...up to the Markov Equivalence class". We will clarify this.
5. Experiments: Comparing with standard MB discovery is interesting but the results will vary depending on underlying structures. For targets with few spouse v-structures, the improvement will be significant. For targets with many spouse v-structures, the improvement will be less significant. Nevertheless in both cases our algorithm improves baseline MB discovery methods. About the efficiency measure, existing local-to-global and MMHC algorithms all invoke MB discovery algorithms, and thus their comparison with our algorithm is fair. For algorithms that do not invoke MB discovery (i.e., P-C and LCD2), we will add the time comparison as suggested. Results from new experiments on the time complexity show that CMB is about 1 to 3 orders of magnitude faster than P-C, about 3 to 8 time faster than LCD2 on different datasets.

R4: thanks for the encourage words.

R5:
1. Contribution vs LCD/BLCD: our algorithm focuses on identifying all causal links (in term of direct causes and effects) with respect to one node, while LCD/BLCD algorithms aim to identify a subset of causal links for all nodes (via special structures) in a causal DAG. Our algorithms is complete in identifying direct causes and effects of one target while LCD/BLCD is not. We will clearly state these contributions in the introduction.
2. Comparison with global algorithms: The reviewer's concern is valid. But since there are no existing comparable local approaches, we had to compare with the global and local-to-global approaches. The empirical and theoretical results show that our algorithm can discover direct causes and effects with the similar accuracy but uses a fraction of time/independence tests.

R6:
Scalability is a general problem for structure learning and causal discovery. Although our algorithm still cannot practically handle millions to billions nodes in the present form, CMB is shown both theoretically and empirically to have better scaling-up capability and better efficiency, up to orders of magnitude, than the existing global algorithms.